# A randomized trial looking at planning prompts to reduce opioid prescribing

**Jason N. Doctor** [1] ✉, **Marcella A. Kelley**[2], **Noah J. Goldstein**[3], **Jonathan Lucas**[4], **Tara Knight**[1] **& Emily P. Stewart**[1]

Prior work has demonstrated that personalized letters are effective at reducing opioid and benzodiazepine prescribing, but it is unclear whether If/when-then planning prompts would enhance this effect. We conducted a decedent-clustered trial which randomized 541 clinicians in Los Angeles County to receive a standard ($n = 284$), or comparator ($n = 257$) version of a letter with If/when-then prompts. We found a significant 12.85% (6.83%, 18.49%) and 8.32% (2.34%, 13.93%) decrease in the primary outcomes morphine (MME) and diazepam milligram equivalents (DME), respectively. This study confirms the benefit of planning prompts, and repeat letter exposure among clinicians with poor patient outcomes. Limitations include lack of generalizability and small sample size. Clinicaltrials.gov registration: NCT03856593.

Despite a national effort to curb opioid addiction and fatal overdoses, excessive opioid prescribing continues to vary substantially over well-defined geographic areas[1]. One promising way to attack this problem involves notifying physicians by mail when one of their patients has a fatal opioid overdose and urging them to reduce opioid prescriptions to safeguard other patients. For example, an initial notification study found the intervention led to an approximately 10% reduction in morphine milligram equivalents (MMEs) dispensed, fewer new patients started on opioids, and a decrease in high-dose opioid prescriptions 1 to 4 months after receipt of the letter[2]. A follow-up study reported positive spillover effects on benzodiazepine prescription practices[3], a demonstration that the letters improved safe prescription practices for a broader range of scheduled drugs. A separate study showed the effects of these notifications on prescription practices persisted for up to 1 year[4]. Notifications may have a lasting impact on patient safety and potentially could reduce the risk of future overdose. Overall, these studies provide evidence that notification of patients' fatal overdose, a low-cost solution that is easily scalable, may have several positive effects.

There might be ways to improve such letters to ensure robust, safe prescription practices. One potential approach is to incorporate planning prompts—sometimes referred to as implementation intentions or If/when-then plans—into these letters[4–6]. Planning prompts are mental rules that describe events meant to bring about concrete actions in specific situations: If (or when) situation S occurs, then engage in behavior B. For example, physicians who receive a notification of a patient's fatal overdose would be prompted to carry out a concrete plan of action that would be triggered by a specific set of circumstances (e.g., when their patients present with pain).

With the incorporation of planning prompts into these letters, physicians may be more likely to take concrete steps to modify their prescription practices and to reduce the risk of future overdose. For example, a physician might better use the information in the letter, if the letter guidance urges them to implement steps at the visit, such as discussing alternative pain management strategies or consulting with a pain management or addiction specialist for evaluation and care.

We hypothesize that the incorporation of planning prompts into notifications will further enhance the effectiveness of the communication, lead to even lower doses and fewer opioid orders, and possibly reduce overdose risk.

To test this hypothesis, we conducted a decedent-clustered randomized trial to compare the effectiveness of standard notification letters to those that incorporate If/when-then plans in Los Angeles County, where the standard letter is mandated by the County Board of Supervisors. We conjecture that If/when-then plain text in the letters will bridge the gap between physicians' intentions and actions; this will result in improved prescription practices.

[1]Sol Price School of Public Policy, University of Southern California, Los Angeles, CA 90089, USA. [2]Edwards Lifesciences, Irvine, CA 92614, USA. [3]UCLA Anderson School of Management, UCLA Geffen School of Medicine, Los Angeles, CA 90095, USA. [4]Department of Medical Examiner-Coroner, County of Los Angeles, Los Angeles, CA, USA. ✉e-mail: jdoctor@usc.edu

The trial's comparison of the effectiveness of a standard notification letter to one that incorporates If/when-then plans will provide insight into not only which letter leads to reduced dose and frequency of opioid orders but will also identify whether certain physicians, such as those who have more patient deaths, benefit the most from the more effective letter. Our trial aims to advance the design of effective interventions to improve patient safety and reduce the risk of future overdose.

## Results

### Opioids sample

Figure 1 shows the sample progression. The Los Angeles Medical Examiner-Coroner examined 316 fatal accidental opioid-related overdoses from late October 2018 to late May 2020 in Los Angeles County. Of these, 236 decedents (74.7%) received at least one CURES-documented scheduled drug from 541 prescribers in the 12 months prior to their death. Thirty-one prescribers (5.73%) had no opioid prescription during the study period and were removed. Three decedents (1.27%) received Scheduled II–IV prescriptions from these prescribers only and were also removed. Since we assigned each prescriber to a cluster by his or her first decedent, this resulted in a final analytic sample of 219 first-decedents and 510 clinicians.

There was no difference in decedent age, sex, race, or cause of death between study arms (Table 1). There was a small difference

($P = 0.025$) in prescriber professional practice between study arms (Table 2). On average, decedents received prescriptions from 2.40 ($\sigma = \pm 2.25$) prescribers, with a range of 1–13. Four hundred sixty-nine prescribers (91.96%) had one decedent. Forty-one prescribers (8.04%) had more than one decedent and received a letter for each. The average number of decedents per prescriber (corresponding to letters sent) was 1.20 ($\sigma = \pm 0.40$). The number of prescribers who had more than one decedent did not differ by study arm in the analytic sample ($\chi^2 = 1.21$, $P = 0.271$).

### Morphine milligram equivalents (MMEs)

Of the 1,538,821 prescriptions dispensed during the study, 559,658 (36.37%) were for an opioid. Logarithm-transformed MME normalized the distribution (Supplementary Fig. S1a, b). Table 3 shows the change in average total weekly MME pre-to-post intervention between study arms. In the comparator arm, the average weekly MME decreased from 157.81 (95% CI: 153.85, 161.76) pre-intervention to 77.05 (95% CI: 75.12, 78.98) post-intervention, compared to 157.70 (95% CI: 153.45, 161.96) and 103.16 (95% CI: 100.34, 105.98) in the standard arm. The difference in average weekly MME pre-to-post intervention was −80.76 (95% CI: −82.92, −78.60) in the comparator arm and −54.55 (95% CI: −56.05, −53.04) in the standard arm. The difference in average weekly MME pre-to-post intervention between study arms was −26.21 (95% CI: −29.63, −22.86), corresponding to a 12.85% (95% CI: 6.83%, 18.49%;

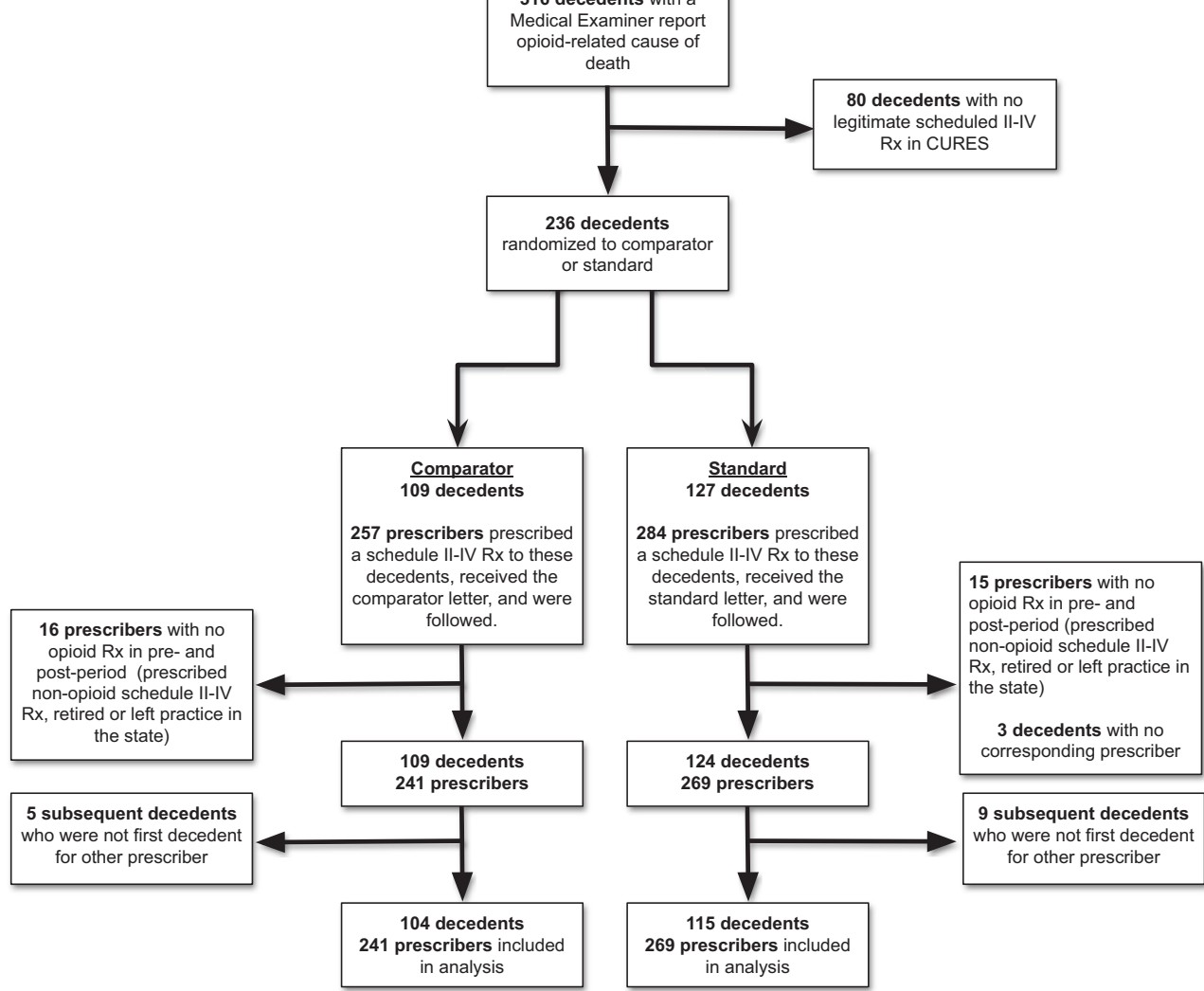

**Fig. 1 | Opioid Prescriber CONSORT diagram.** CONSORT diagram for clinicians who prescribed an opioid during the study period.

**Table 1 | Decedent characteristics[1]**

| Randomization group | | | | |
|---|---|---|---|---|
| Characteristic[a] | Comparator (N = 109) | Standard (N = 127) | Statistic[e] | Two-sided P value |
| Age, (±σ) | 40.76 (13.13) | 40.02 (13.92) | $t = -0.42$ | 0.674 |
| **Gender[b]** | | | | |
| Female | 32 (29.36%) | 29 (22.83%) | | |
| Male | 77 (70.64%) | 98 (76.38%) | $\chi^2 = 1.30$ | 0.253 |
| **Race** | | | | |
| Hispanic | 33 (30.28%) | 26 (20.47%) | | |
| White | 61 (55.96%) | 80 (62.99%) | $\chi^2 = 3.04$ | 0.219 |
| Other/missing | 15 (13.76%) | 21 (16.54%) | | |
| **Cause of death** | | | | |
| Opioid[c] prescription only | 37 (33.94%) | 34 (26.77%) | | |
| Heroin only | 13 (11.93%) | 25 (19.69%) | | |
| Opioid prescription and heroin | 49 (44.95%) | 48 (37.8%) | $\chi^2 = 5.92$ | 0.116 |
| Other/missing[d] | 10 (9.17%) | 20 (15.75%) | | |

OTC over the counter, n number of clinicians.

[a]Counts of less than 10 were censored in accordance with the State of California Department of Justice's CURES policy.

[b]One decedent's missing gender was deduced using first name.

[c]Supplemental Table 4a contains opioid types.

[d]Other are multi-causal deaths that in addition to opioids, included alcohol/ethanol, non-opioid Schedule II–IV prescriptions (e.g., benzodiazepines, anticonvulsants, muscle relaxants, and barbiturates), and/or illicit drugs (e.g., methamphetamine, cocaine, MDMA, and ketamine).

[e]Two-sample t-test for continuous traits. Chi-square test for categorical variables.

**Table 2 | Prescriber characteristics**

| Characteristic[a] | Randomization group | | Statistic[b] | Two-sided P value |
|---|---|---|---|---|
| | Comparator (n = 257) | Standard (n = 284) | | |
| **Gender** | | | | |
| Male | 99 (38.52%) | 104 (36.62%) | | |
| Female | 27 (10.51%) | 29 (10.21%) | | |
| Declined to disclose | 58 (22.57%) | 67 (23.59%) | $\chi^2 = 0.27$ | 0.966 |
| Missing | 73 (28.40%) | 84 (29.58%) | | |
| **Professional practice** | | | | |
| Medical doctor (MD) | 172 (66.93%) | 194 (68.31%) | | |
| Doctor of osteopathy (DO) | 20 (7.78%) | 12 (4.23%) | | |
| Nursing (NP/FNP/DNP) | 23 (8.95%) | 39 (13.73%) | $\chi^2 = 11.13$ | 0.025 |
| Physician assistant (PA) | 20 (7.78%) | 28 (9.86%) | | |
| Other | 22 (8.56%) | 11 (3.87%) | | |
| **Primary specialty** | | | | |
| Emergency medicine | 31 (12.06%) | 37 (13.03%) | | |
| Internal medicine | 44 (17.12%) | 43 (15.14%) | | |
| Psychiatry | 21 (8.17%) | 22 (7.75%) | | |
| Family medicine | 22 (8.56%) | 20 (7.04%) | $\chi^2 = 1.18$ | 0.947 |
| Other | 41 (15.95%) | 45 (15.85%) | | |
| Missing | 98 (38.13%) | 117 (41.20%) | | |
| **Race/Ethnicity** | | | | |
| Asian/Pacific Islander | 21 (8.17%) | 14 (4.93%) | | |
| Non-Hispanic White | 18 (7.00%) | 20 (7.04%) | | |
| Other | 12 (4.67%) | 23 (8.10%) | $\chi^2 = 5.02$ | 0.285 |
| Declined to disclose | 132 (51.36%) | 139 (48.94%) | | |
| Missing | 74 (28.79%) | 88 (30.99%) | | |
| *Location* | | | | |
| Los Angeles City | 240 (93.39%) | 259 (91.20%) | | |
| Long Beach or Pasadena City | 17 (6.61%) | 25 (8.80%) | $\chi^2 = 0.902$ | 0.342 |
| **No. of decedents** | | | | |
| >1 | 16 (6.23%) | 25 (8.80%) | | |
| 1 | 241 (93.77%) | 259 (91.20%) | $\chi^2 = 1.28$ | 0.258 |
| No. of decedents, (±σ) | 1.07 (0.27) | 1.10 (0.32) | $t = 1.16$ | 0.248 |

[a]Counts of less than 10 were censored in accordance with the State of California Department of Justice's CURES policy.

[b]Two-sample t-test for continuous traits. Chi-square test for categorical variables.

$P < 0.001$) greater decrease in MME among prescribers randomized to the comparator letter (Supplementary Table S1a).

To evaluate if outlier clinicians played a role in study effects, Fig. 2 shows the clinician-level total log MME distribution pre-to-post intervention between study arms among clinicians with one decedent and more than one decedent. Pre-intervention, the dependent variable did not differ by study arm ($t = 0.83$, $P = 0.406$), or among clinicians with one versus multiple decedents ($t = -0.61$, $P = 0.544$). The number of outliers also did not differ by study arm ($\chi^2 = 1.42$, $P = 0.233$) or number of decedents ($\chi^2 = 0.48$, $P = 0.759$) (Supplementary Table S2a).

### Exploratory opioid outcomes

Prescribers in the comparator arm were less likely to prescribe an opioid to a new patient post-intervention, but not significantly ($\beta = -0.09$; [95% CI: −0.19, 0.01]; $P = 0.07$). There was also no significant difference in the odds of a patient receiving a prescription of at least 50 MME ($\beta = -0.01$; [95% CI: −0.12, 0.11]; $P = 0.894$). Clinicians randomized to the comparator letter had significantly lower odds ($\beta = -0.18$; [95% CI: −0.34, −0.02]; $P = 0.027$) of a patient receiving a medication greater than 90 MME compared to those who received the standard letter.

### Post-hoc morphine milligram equivalent analyses

There was no difference in the study-start and study-end coefficients ($\beta = -0.06$ [95% CI: −0.16, 0.04]; $P = 0.224$), indicating that the effect persisted over time. There was a significant three-way interaction between study arm, time, and number of decedents, translating to a 31.41% ([95% CI: 11.38%, 46.91%]; $P = 0.004$) greater decrease in total weekly MME for prescribers who received multiple comparator letters (Supplementary Table S3a).

### Benzodiazepine sample

Thirteen clinicians (2.40%) did not prescribe a benzodiazepine during the study period and were removed from the analysis (Fig. 3). One decedent (0.42%) received scheduled prescriptions from these prescribers and was also removed. The final analytic sample was 528 clinicians and 220 first-decedents.

Relative to opioid prescribers, decedents received scheduled prescriptions from a slightly higher number of clinicians who prescribed a benzodiazepine ($X = 2.44$, $\sigma = \pm 2.27$; range: 1–13). Benzodiazepine prescribers received an average of 1.21 ($\sigma = \pm 0.42$) letters. Forty-one (7.77%) prescribers had more than one decedent. The number of decedents did not vary by study arm ($\chi^2 = 1.29$, $P = 0.256$).

### Diazepam milligram equivalents (DME)

Three hundred eighty-four thousand seven hundred sixty-eight (25%) dispensed scheduled prescriptions for a benzodiazepine. Log transforming DME normalized the distribution (Supplementary Fig. S1c, d).

Average weekly DME decreased from 51.47 (95% CI: 49.83, 53.11) pre-intervention to 30.16 (95% CI: 29.27, 31.05) post-intervention in the comparator arm and from 54.36 (95% CI: 52.90, 55.82) to 43.71 (95% CI: 42.49, 44.92) in the standard arm (Table 4). The difference in pre-to-post-intervention average weekly DME was −21.31 (95% CI: −21.98, −20.64) in the comparator arm and −10.65 (95% CI: −10.95, −10.35) in the standard arm. The difference-in-difference in average weekly DME was −10.66 (95% CI: −12.27, −9.04). This corresponds to an 8.32% ([95% CI: 2.34, 13.93]; $P < 0.01$) greater decrease in DME (Supplementary Table S1b).

To evaluate if outliers contributed to the effect, Fig. 4 shows the pre-to-post intervention clinician-level logarithm transformed DME distribution by study arm and number of decedents. There were no differences by study arm in log DME ($t = 1.50$, $P = 0.133$). Pre-intervention log DME was slightly higher among clinicians with more than one decedent ($t = −5.28$, $P < 0.001$). The number of outliers also did not differ by study arm ($\chi^2 = 3.65$, $P = 0.125$) or number of decedents ($\chi^2 = 0.09$, $P = 1.0$) (Supplementary Table S2b).

**Table 3 | Adjusted mean total weekly morphine milligram equivalents (MMEs) dispensed between study arms. Values in parentheses are 95% CIs with 9% trimmed means**

| Parameter | Randomization group | |
|---|---|---|
| | Comparator | Standard |
| Prescribers followed | 241 | 269 |
| Pre-intervention | 157.81 (153.85, 161.76) | 157.70 (153.45, 161.96) |
| Post-intervention[a] | 77.05 (75.12, 78.98) | 103.16 (100.34, 105.98) |
| Increment (pre- to post-) | −80.76 (−82.92, −78.60) | −54.56 (−56.05, −53.04) |
| Difference in increment | −26.21 (−29.63, −22.86) | |
| Two-sided P value | <0.001 | |

[a]Predicted MME using coefficients from censored, mixed linear model (Table S1a) testing the two-sided hypothesis that change in pre-to-post MME does not differ by study arm.

### Exploratory benzodiazepine outcomes

Prescribers in the comparator arm were not less likely to prescribe a benzodiazepine to a new patient post-intervention ($\beta = 0$; [95% CI: −0.09, 0.09]; $P = 0.962$). Comparator arm clinicians did not have higher odds of reducing DME by more than 20% ($\beta = −0.14$; [95% CI: −0.54, 0.26]; $P = 0.488$).

### Post-hoc diazepam milligram equivalent analyses

There was no difference in the study-start and study-end coefficients ($\beta = −0.06$ [95% CI: −0.15, 0.04]; $P = 0.229$), indicating that the effect persisted over time. The three-way interaction between study arm, time, and number of decedents was significant (Supplementary Table S3b); prescribers who received multiple comparator letters had a 56.05% ([95% CI: 45.04%, 64.85%]; $P < 0.001$) greater reduction in total weekly DME.

### Discussion

We sought to determine whether the addition of If/when-then plans could bridge the gap between physicians' intentions and actions in notifications of a fatal overdose in a physician's practice. Our findings suggest that If/when-then plans may reduce opioid prescription intensity and frequency, thereby reducing future risk. A moderator analysis showed that the If/when-then plan was more effective relative to the standard letter for prescribers who received multiple letters. These were persons with multiple deaths in their practice. There was also evidence of spillover to more judicious benzodiazepine prescribing. A safety analysis showed that study arms did not differ in the number of clinicians demonstrating a 20% reduction in filled benzodiazepine prescriptions. The If/when-then plan may have helped prescribers with complex or difficult patients to implement skills that promote safe prescription practices. Alternatively, the prescribers with multiple deaths may have lacked skills and benefited from a letter that carefully outlined steps to improve prescription safety.

The intervention is scalable. Forty-nine of the 50 U.S. states operate prescription drug monitoring programs, and every county in the U.S. has a medical examiner or coroner. These interventions could

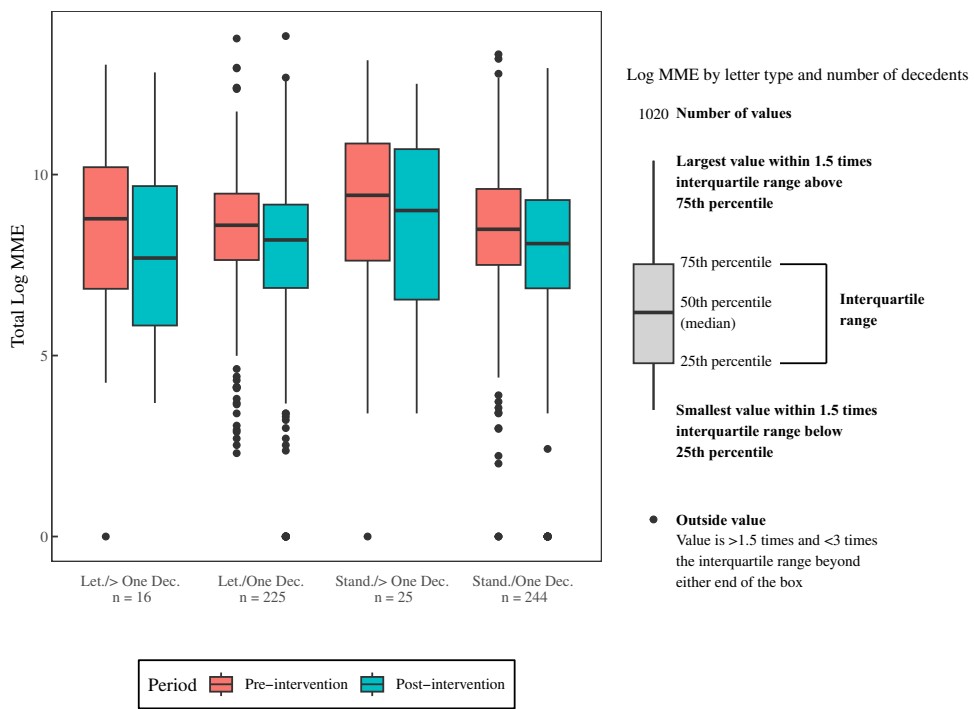

**Fig. 2 | Clinician-level Opioid Prescribing.** Boxplots showing median log MME, the interquartile range, and outliers pre-to-post intervention between study arms among clinicians with one versus multiple decedents.

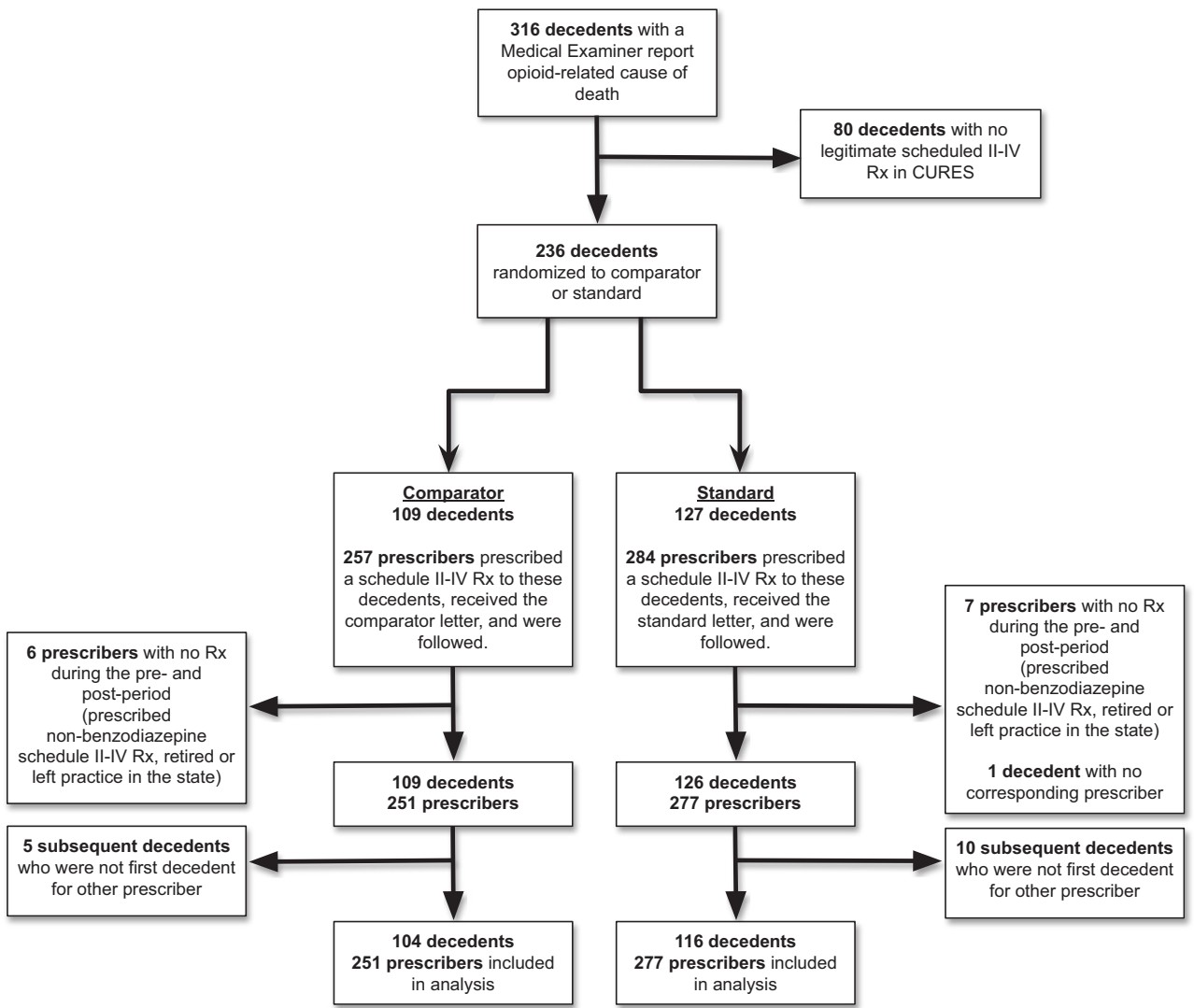

**Fig. 3 | Benzodiazepine Prescriber CONSORT diagram.** CONSORT diagram for clinicians who prescribed a benzodiazepine during the study period.

easily operate out of the medical examiner's office almost anywhere in the U.S. This intervention has the benefit of helping clinicians learn of deaths in their practice that they might not otherwise learn about. This creates a more balanced sample from which they can draw inferences about their patients' outcomes. The intervention is low-cost and requires only minimal changes to routine administrative tasks in the medical examiner's office. A downside to the intervention is that it cannot reach all clinicians, but only those with a death in their practice.

### Table 4 | Adjusted mean total weekly diazepam milligram equivalents (DMEs) dispensed between study arms. Values in parentheses are 95% CIs with 7% trimmed means

| Parameter | Randomization group | |
|---|---|---|
| | **Comparator** | **Standard** |
| Prescribers followed | 251 | 277 |
| Pre-intervention | 51.47 (49.83, 53.11) | 54.36 (52.90, 55.82) |
| Post-intervention[a] | 30.16 (29.27, 31.05) | 43.71 (42.49, 44.92) |
| Increment (pre- to post-) | −21.31 (−21.98, −20.64) | −10.65 (−10.95, −10.35) |
| Difference in increment | −10.66 (−12.27, −9.04) | |
| Two-sided P value | 0.007 | |

[a]Predicted DME using coefficients from censored, mixed linear model (Table S1b) testing two-sided hypothesis that change in pre-to-post DME does not differ by study arm.

Those clinicians with a death in their practice are, however, in greatest need of receiving the intervention.

Doctor et al. [2] compared a letter notifying clinicians of a death in their practice to a no-treatment control group. Morphine milligram equivalents in prescriptions filled by patients of letter recipients versus no intervention controls decreased by 9.7% (95% CI: 6.2 to 13.2%; $P < 0.001$)[2]. The current paper compared a modified letter to the active treatment in Doctor et al. 2018, which resulted in a 12.9% decrease in morphine milligram equivalent prescriptions filled. The new letter appears even more effective than the letter in Doctor et al. (2018). From the standpoint of psychological mechanisms, the plan may have provided a simple way to take contingent action at pain-related visits. Our results are in line with previous research that shows If/when-then plans are effective in other environments[7–9].

This study has several strengths. It is a randomized concealment design[10] that avoids standard pitfalls that come with trial enrollment, attrition, and non-response. The study also used unobtrusive measures (California's Controlled Substance Utilization Review and Evaluation System 2.0 [CURES]), which lowered the potential for reactivity. Despite the strengths of our study, there are several limitations to consider. First, our study was conducted in Los Angeles County, and while it is the largest County in the United States, the results may not be generalizable to other populations or geographic regions. Additionally, our sample size was limited, and our study only examined the

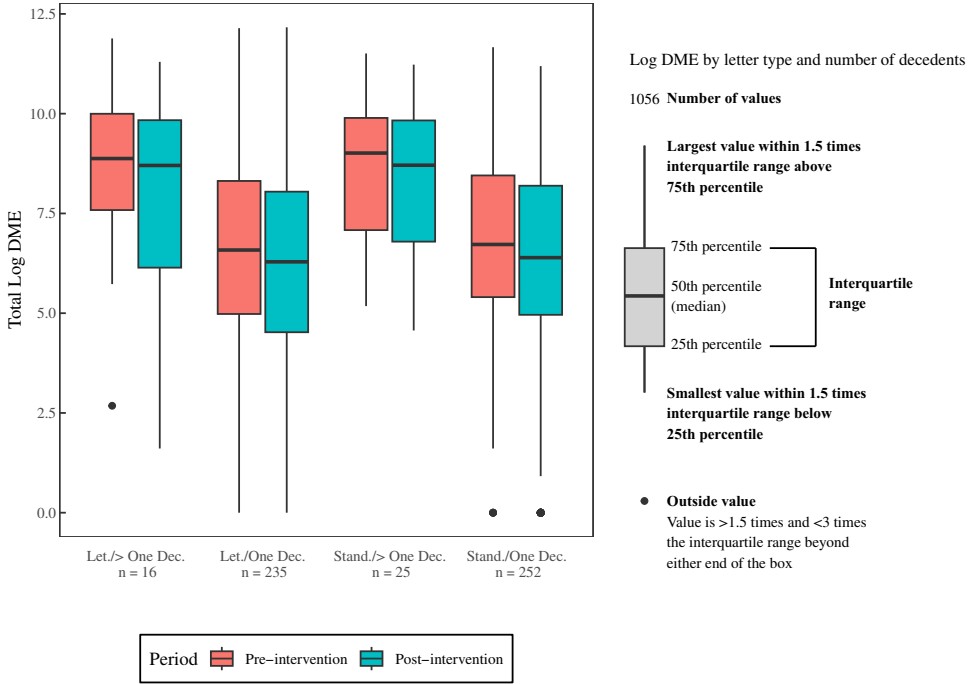

**Fig. 4 | Clinician-level Benzodiazepine Prescribing.** Boxplots showing median log DME, the interquartile range, and outliers pre-to-post intervention between study arms among clinicians with one versus multiple decedents.

short-term effects of the intervention. Finally, we were unable to examine the effects of the intervention on patient outcomes, such as overdose rates or other adverse events.

In conclusion, our study provides evidence that If/when-then plans as additions to fatal overdose notification letters moderate physician prescription practices. Further research is needed to confirm the results, examine the long-term effects of the intervention, and explore its potential effects on patient outcomes.

## Methods

The trial protocol contains details regarding the intervention, power calculations, randomization, inclusion criteria, and statistical approach (see Supplement). All study procedures were ethically compliant and reviewed and approved by the University of Southern California's Institutional Review Board (IRB) prior to trial implementation (UP-19-00172). As part of this approval, we were granted a waiver of informed consent and a full waiver of HIPAA authorization. Participants were not compensated.

### Intervention

This cluster randomized controlled trial compares two versions of a personal notification of a patient's fatal overdose designed to reduce Schedule II-IV prescriptions and opioid dosages. Trial participants were Schedule II-IV prescribers to individuals who died of an overdose between late October 18, 2018 and late May 21, 2020 in Los Angeles County, where opioids were a contributing or primary cause. We received decedent demographics and toxicology information from the Los Angeles County Department of Medical Examiner-Coroner. The contributing cause of death was determined by the Medical Examiner's judgment. The decedent's gender was obtained from a government-issued ID and, therefore, self-reported. Clinician gender was also self-reported, and retrieved via license search on the Department of Consumer Affairs site[11]. Gender was not considered in the study design. We received a waiver of consent, which does not allow us to report disaggregated, individual-level data for gender, or any other demographic characteristic. The Medical Examiner staff downloaded prescriber names for each decedent through the State of California

Department of Justice's CURES online portal[12]; data was maintained on a secure, electronic database with restricted access by the Los Angeles Department of Medical Examiner-Coroner. Prescribers were grouped into decedent clusters and randomized to receive a standard letter or comparator.

The standard letter (see Supplementary Methods S1) was used in Doctor et al.[2]. Both letters informed prescribers of their patient's opioid-related overdose, provided guidance on judicious prescription practices, and directed the prescriber to CURES. Both letters were also signed by the Los Angeles County Chief Medical Examiner-Coroner and one of the Los Angeles, Pasadena, or Long Beach Public Health Officers as determined by the prescriber's practice location. The comparator letter, however, included an If/when-then plan that advised the clinician to keep the letter's recommendations close when their next patient presents with pain (see Supplementary Methods S2). The If/when-then plan specifies a contingent action. Letters were sent to prescribers on a monthly basis between April 4, 2019 and July 8, 2020.

California pharmacies that dispense controlled substances are legally required to submit weekly reports to CURES. We accessed this data in collaboration with the research unit and program manager of CURES; per decedent, we notified administrators that clinician prescriptions would be prospectively analyzed post-randomization and letter receipt. We provided CURES with prescriber names to retrieve all Schedule II-IV prescriptions from October 1, 2017 to August 31, 2021. CURES replaced personal identifiers with randomized digit IDs for transfer to the University of Southern California.

### Power calculations

Power was calculated with the statistical computing language R clusterPower package. We assumed a mean of 5.5 prescribers per decedent[2], and a coefficient of variation of 1.22. This limited the number of prescribers per decedent at the 99th percentile to 20. Relative to the intraclass correlation of 0.05–0.15[13] found for most clinician process measures, we used a more conservative estimate of 0.20. Using the Taylor method for variance inflation due to unequal clusters and given a two-tailed test with a 5% Type 1 error rate, we had

80% power to detect a 50% reduction in mean MME or DME ($\sigma = \pm 140$) with 103 decedents per study arm.

## Inclusion and exclusion criteria

**Decedents.** Decedents were included in the randomization if they died of an opioid-related overdose in Los Angeles County from October 18, 2018 to May 21, 2020, and received a legitimate Schedule II–IV prescription from a clinician verified by CURES within the 12 months prior to their death.

**Prescribers.** Prescribers were included if they (1) were located and practicing in LA County, (2) had scheduled drug prescribing privileges and, (3) prescribed a Schedule II–IV drug within the 12 months prior to a patient's death where opioids were the primary or a contributing cause.

## Randomization

Decedents who died of an opioid-related overdose in Los Angeles County from October 18, 2018 to May 21, 2020, and received a CURES-verified Schedule II-IV prescription, were randomized to the comparator or standard letter. Decedent randomization was stratified by whether the decedent was prescribed a benzodiazepine in the year prior to death. For each of the 2 combinations of stratification levels, the principal investigator and lead analyst created a list of decedents. We used random.org's list randomizer to randomize decedent clusters to a numbered list. The first half of the list was randomized to the standard arm, and the second half of the list was randomized to the comparator arm. If the number of decedents was not divisible by 2, the last decedents on the randomly numbered list were assigned by a ½ probability lottery on random.org. The randomization process was repeated for new prescribers each monthly cycle. Prescribers with multiple decedents received multiple letters. Their experimental condition did not change after the first letter, ensuring they received the same letter repeatedly.

## Measures

In addition to opioids, we analyzed benzodiazepines for potential spillover effects. Opioid and benzodiazepine prescriptions dispensed in the 52 weeks prior to a prescriber's letter sent date were included in the pre-intervention period. Opioid and benzodiazepine prescriptions dispensed 4–52 weeks after the letter sent date were included in the post-intervention period. The first 4 weeks of follow-up were washed out to avoid contamination from undispensed prescriptions ordered before letter receipt. In the case of prescribers with multiple decedents, the first decedent's letter sent date defined the pre- and post-intervention periods.

## Outcomes

We used the Centers for Disease Control for MME conversion factors and guidelines (Supplemental Table S4a)[14]. We converted benzodiazepines to diazepam milligram equivalents (DME) based on information provided by Borrelli et al.[15]. (Supplemental Table S4b). Our primary outcomes were the change in total weekly MME and DME dispensed pre- to post-intervention between study arms. Per-prescription average daily MME and DME were calculated by multiplying the prescription strength by the number of units prescribed per day and the conversion factor (i.e., strength*(quantity/days)*conversion factor)[15]. This was summed per clinician per week.

We log-transformed total weekly MME and DME to ensure normality. We confirmed normality with quantile-quantile (q-q) plots. Pre-intervention outliers were detected using Tukey's fences[16], which is 1.5 times the interquartile range plus or minus quartiles three and one, respectively.

Our exploratory outcomes were the difference in the likelihood of an opioid prescription greater than or equal to 50 MME and the

difference in the likelihood of an opioid prescription greater than 90 MME. We also tested the difference in the probability of a new patient receiving an opioid or benzodiazepine prescription pre- to post-intervention between study arms. To test patient safety associated with spillover effects, we assessed pre-to-post-intervention mean DME drop-offs greater than 20%.

## Statistical analysis

Statistical analyses were executed using SAS version 9.4, STATA software version 16, and R version 4.3.2[17]. We used a multilevel regression with left censoring to account for observations without any opioid or benzodiazepine prescriptions[18]. We regressed log MME and DME on the study arm (comparator vs. standard letter), the study period (pre-intervention vs. post-intervention), and the interaction between the study arm and the study period. To control for differences in decedent and prescriber behavior, we included a random intercept for the prescriber nested within the decedent. Model 1 is the effect of time ($\beta_1$) study arm ($\beta_2$), and the interaction between time and study arm ($\beta_3$) on $Y$ for the $ith$ prescriber and $jth$ prescription, where $Y$ is uncensored, latent weekly log MME or log DME. The coefficients are a $[100*(1 - \exp(\beta)]$ change in uncensored MME and DME per-level increase. $\delta_{i(k)}$ is the random intercept for prescriber nested within the decedent, with mean 0 and variance $\sigma_{i(k)}$. Due to prescriber outliers in the standard letter group, we calculated trimmed pre-intervention means for Table 3 and Table 4 that resulted in the smallest difference between letter groups, which was 9% and 7% for opioid and benzodiazepine prescribers, respectively. We calculated adjusted post-intervention means to obtain the difference-in-difference in pre- to post-weekly MME and DME between the standard and comparator study arms. 95% confidence intervals were bootstrapped using 2000 repetitions of 12,000 randomly selected observations.

$$\log(\hat{Y})^* = \beta_1 x_{1ij} + \beta_2 x_{2ij} + \beta_3 x_{3ij} + \delta_{i(k)}$$

We evaluated exploratory outcomes using mixed-effects, logistic regressions to test whether patients with prescribers randomized to the comparator letter were less likely to receive an opioid prescription greater than or equal to 50 MME or greater than 90 MME post-intervention and the probability of a new patient receiving an opioid or benzodiazepine prescription post-intervention. We calculated the per-clinician percent change in mean DME pre-to-post intervention and used logistic regression to test whether greater than 20% drop-offs were higher among clinicians in the comparator arm, controlling for the proportion of new users and opioid coprescriptions.

We conducted two post hoc analyses. The first added a three-way interaction between study arm, time, and number of decedents to Model 1 to test whether high-frequency prescribers with multiple decedents were more amenable to repeat comparator letter exposure. The second assessed whether letter efficacy decreased over time by including two fixed interaction terms between the study arm, the study start (weeks 4–22), and the study end (weeks 23–52). We compared study-start and study-end coefficients using an equality of regression coefficients test[19]. Boxplots and corresponding legends were generated using the R officer, cowplot, rvg, and ggplot2 packages[20].

## Reporting summary

Further information on research design is available in the Nature Portfolio Reporting Summary linked to this article.

# Data availability

The datasets generated and/or analyzed during the current study involve third-party data from the California Department of Justice and are not publicly available as they contain protected health information, posing participant confidentiality and privacy concerns. Data may be available jointly through the Department of Justice and the

corresponding author (J.N.D.) through a signed Data Use Agreement. Requests should be submitted to jdoctor@usc.edu; allow 30 days for a response to your request.

## Code availability

All code used for data management, descriptive analyses, model fitting, and plotting is publicly available on a GitHub repository at LA-Letters-Code/ at main · epstewart111/LA-Letters-Code · GitHub. We have also used Zenodo to assign a DOI to the repository: 10.5281/zenodo.10263890. The license used to generate the code is the Schaeffer Center for Health Policy and Economics, University of Southern California.

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

## Acknowledgements

This study was supported by the National Institutes on Aging (P30AG024968; PI: Dr. Jason Doctor, University of Southern California). The funding agency had no role in the design and conduct of the study; collection, management, analysis, or interpretation of the data; preparation, review, or approval of the paper; or decision to submit the paper for publication.

## Author contributions

M.K. and E.P.S. had full access to all of the data in the study and took responsibility for the integrity of the data and the accuracy of the data analysis. Concept and design: J.N.D., N.J.G and J.L. Acquisition, analysis, or interpretation of data: J.N.D., M.K., N.J.G., J.L., T.K.K. and E.P.S. Drafting of the paper: J.N.D., M.K., E.P.S. and N.J.G. Critical revision of the manuscript for important intellectual content: J.N.D., M.K., N.J.G., J.L., T.K.K. and E.P.S. Statistical analysis: M.K., E.P.S. and J.N.D. Obtained funding: J.N.D. Administrative, technical, or material support: T.K. and J.N.D. Supervision: T.K. and J.N.D.

## Competing interests

The authors declare no competing interests.
