## [Peer Review File · Nature Communications]

A randomized trial looking at planning prompts to reduce opioid prescribingREVIEWER COMMENTS

Reviewer #1 (Remarks to the Author):

The authors produced a nicely written manuscript describing a randomized controlled trial examining how letters that incorporated planning promos outperformed a standard letter in facilitating a reduction in morphine equivalents. This was particularly true of clinicians who received multiple letters due to having had multiple patients die from opioid overdose.

Strengths of this manuscript is that it was well-written, thorough. It satisfied all 25-items on the CONSORT checklist. More importantly, there is an intervention with a notable impact in the community. I appreciate that the authors anticipated my question in their post-hoc analysis: namely, the persistence of the effect.

The tested intervention is actionable and available to public health officials.

The analysis was thorough and transparent and no significant flaw revealed itself to this reviewer.

One potential area to improve the impact:

Please include a copy of the If/when-then plan letter, highlighting the critical language. Your study argues that the health community can be better informed by such language. It feels necessary to include that language if you truly intend to amend practice. To that end, the authors should include a copy of the standard letter. Yes it is available in the supplement of the author's Science paper, but it does not facilitate seeing the impact of the comparator.

Reviewer #2 (Remarks to the Author):

Overall this is a well written paper that demonstrates a project that does meet the goal set out of decreasing opioid prescriptions after the intervention. This paper is worthy of strong editorial consideration to publish.

It would be stronger if there was an assessment of pain management practices and the impact on patients in these practice, but the project was not designed to test that. There are

patients that benefit from managed opiates for pain and simply reducing the opioid equivalents prescribed is not a patient centered outcome. Some patients may also benefit from medication assisted therapy for opioid use disorder. Communication strategies like the intervention in this paper, does highlight the risk of overdose, and is a valuable part of larger intervention.

Few edits. (none of my copies had page or line numbers so I counted lines)

Introduction, page 1 of intro:

Line 18, Delete "Yet" start line with "There"

Page 2 of introduction:

Line 22, Delete "Overall" start sentence with "Our"

Methods: Excellent methods to answer the narrow question that is defined. Cluster randomization is the correct tool for this type of trial.

Results: No edits, analysis and study design are reasonable and well done.

Discussion: Concise, well written, summarizes what is literature there is on this topic and the importance of the project.

Reference; Appropriate to project.

Reviewer #3 (Remarks to the Author):

Thank you for the opportunity to review this interesting article by Doctor and colleagues. In this study, they build on their groups prior work using notification letters and other behavioral techniques to promote high quality prescribing practices by physicians. This trial randomized 510 physicians to receive either a notification letter after a prescribed patients'

overdose following the method in Doctor et al 2018, or an "enhanced" letter with a contingency plan included to help provide concrete next steps for clinicians to follow. The authors find that the "enhanced" letters prescribed fewer opioids after intervention exposure, an effect driven by doctors who received multiple letters due to multiple decedents in their prescribing pool.

Overall, I think this paper is valuable though I have important concerns below. I think this result is quite interesting and potentially actionable for future letter interventions. The authors took a big risk in comparing one form of a letter to another form, but it seems that we can tweak the format to get better effectiveness.

Here are my major concerns:

1) Unless I am missing it somehow, the paper needs to include examples of the letters in the two trial arms. The description in the paper is not detailed enough to really understand what clinicians saw. This makes it hard for me to have a clear understanding of how to interpret the results. It's possible that the format of the letter might be a bit more heavy handed or have other features that might potentially influence responsiveness.

2) The effect is so exaggerated in the small (<10%) >1 decedent subgroup that it seems like something odd is going on in the data that the mean effects are obscuring. The confusing wording in the methods of how extreme outliers were censored adds further concern that the mean effect may be the result of a very small number of physicians. ("Due to prescriber outliers in the standard letter group, we calculated trimmed pre-intervention means that resulted in the smallest difference between letter groups. This was 9%, which we trimmed observed pre-intervention means by.")

One way to address this would be to include a waterfall plot that shows the entire distribution of physician-level changes pre/post, so that readers can assess the difference in distribution themselves. Another would be to provide more transparency on the magnitude of outliers and the distribution of prescribing patterns across all participants in both arms.

Minor issue on this point: the 1 vs. >1 decedent characteristic should be reported in Table 2.

3) The conclusions is remarkably short for such a large effort devoted to making this analysis happen. It basically just restates the main result without very little interpretation. Here are examples of questions I would have liked to see the authors examine in the conclusion:

- What could explain the 1 vs. >1 decedent subgroup analyses?
- Could this intervention be scalable? Would that be a good idea?
- Are there any potential drawbacks to this as a public health strategy?
- How do these results compare to Doctor et al 2018 and other papers in the literature?

This is just a partial list off the top of my head. The conclusion should be 2-3x more detailed than it is.

More minor concerns:

- 1) How do you know if the cause of death is "illicit" or "prescribed" opioids? These categories are not defined.
- 2) Seems odd that you only have white/hispanic/other race categories. Are there no black decedents in LA County from opioid overdoses in this period? Seems unlikely. Also it's not clear how race is defined since decedents can't self report.
- 3) The writing is very odd in parts of the paper. It really threw me that there are multiple instances of a patient "dispensing" a prescription which is not correct usage.

Response to Reviewer Comments

Reviewer #1 (Remarks to the Author):

Please include a copy of the If/when-then plan letter, highlighting the critical language. Your study argues that the health community can be better informed by such language. It feels necessary to include that language if you truly intend to amend practice. To that end, the authors should include a copy of the standard letter. Yes it is available in the supplement of the author's Science paper, but it does not facilitate seeing the impact of the comparator.

Response: Thank you. We have added both letters to the Supplement (Supplementary Methods S1 and S2), and highlight the “if/when-then” language in the comparator letter (S2).

Reviewer #2 (Remarks to the Author):

Introduction, page 1 of intro:

Line 18, Delete "Yet" start line with "There"

Response: Thank you. We have removed “Yet”.

Page 2 of introduction:

Line 22, Delete "Overall" start sentence with "Our"

Response: Thank you. We have removed “Overall”.

Reviewer #3 (Remarks to the Author):

1) Unless I am missing it somehow, the paper needs to include examples of the letters in the two trial arms. The description in the paper is not detailed enough to really understand what clinicians saw. This makes it hard for me to have a clear understanding of how to interpret the results. It's possible that the format of the letter might be a bit more heavy handed or have other features that might potentially influence responsiveness.

Response: We agree that visually comparing the letters is useful, and have added both to the supplement, highlighting the “if-when/then” language in the comparator (See Supplementary Methods S1 Standard Letter and S2 Comparator Letter).

2) The effect is so exaggerated in the small (<10%) >1 decedent subgroup that it seems like something odd is going on in the data that the mean effects are obscuring. The confusing wording in the methods of how extreme outliers were censored adds further concern that the mean effect may be the result of a very small number of physicians. ("Due to prescriber outliers in the standard letter group, we calculated trimmed pre-intervention means that resulted in the smallest difference between letter groups. This was 9%, which we trimmed observed pre-

intervention means by.")

One way to address this would be to include a waterfall plot that shows the entire distribution of physician-level changes pre/post, so that readers can assess the difference in distribution themselves. Another would be to provide more transparency on the magnitude of outliers and the distribution of prescribing patterns across all participants in both arms.

Response: Thank you for your comment. Outliers were not censored for model coefficients. We trimmed only pre-intervention Table 3 and Table 4 means. We clarify by adding "Table 3 and Table 4" to sentence eight under "Statistical Analysis" in the "Methods".

We log transformed MME and DME to normalize the data used for regression. We have added quantile-quantile (q-q) plots to the Supplement (Fig 1a and 1b) comparing raw versus log MME and log DME. Effects on the log scale correspond to large percentage changes; for example, estimated change in post-intervention log MME for prescribers in the standard condition with multiple decedents is 0.56, translating to a 75% MME increase. The sizable difference in predicted, post-intervention MME between study arms is visible in our original figure. We have since replaced this figure with box plots (Figure 2) showing pre-to-post change in observed (i.e., unadjusted) per-clinician total log MME between study arms, and clinicians with one versus multiple decedents. Although smaller, there is still a noticeable difference in post-intervention MME between study arms on the log scale among clinicians with more than one decedent.

We also assessed pre-intervention, per-clinician total MME and DME outliers and means between study arms, and clinicians with one versus multiple decedents. We have added this to sentences 6-8 under "Outcomes" in the "Methods" section, and report the results in paragraph two under "Primary outcome" in the "Opioids" and "Benzodiazepine spillover effects" subsections of "Results", and in Supplemental Tables S1a and S1b.

Minor issue on this point: the 1 vs. >1 decedent characteristic should be reported in Table 2.

Response: We have added the number of clinicians with one vs. more than one decedent, as well as the mean (SD) number of decedents to Table 2.

3) The conclusion is remarkably short for such a large effort devoted to making this analysis happen. It basically just restates the main result without very little interpretation. Here are examples of questions I would have liked to see the authors examine in the conclusion:

- What could explain the 1 vs. >1 decedent subgroup analyses?*
- Could this intervention be scalable? Would that be a good idea?*
- Are there any potential drawbacks to this as a public health strategy?*
- How do these results compare to Doctor et al 2018 and other papers in the literature?*

This is just a partial list off the top of my head. The conclusion should be 2-3x more detailed than it is.

Response: We agree with the reviewer and have lengthened the Discussion section to include the points raised by the reviewer.

1) How do you know if the cause of death is "illicit" or "prescribed" opioids? These categories are not defined.

Response: We agree our previous wording was vague, and have changed Table 1 “Cause of death” row titles to be more clear. We have also added superscripts where relevant.

The Los Angeles Medical Examiner-Coroner provided a list of all substances found in each decedent based on toxicology reports. We grouped decedents by predefined categories that, in accordance with the State of California Department of Justice’s CURES censoring policy, resulted in cell sizes of ten or more. We state where we received decedent cause of death in sentences 3-4 under “Intervention” in the “Methods” section.

2) Seems odd that you only have white/hispanic/other race categories. Are there no black decedents in LA County from opioid overdoses in this period? Seems unlikely. Also it's not clear how race is defined since decedents can't self-report.

Response: Thank you. There were fewer than ten Black and Asian decedents. Therefore, they were combined with ‘Other/Missing’ in accordance with the State of California Department of Justice’s CURES censoring policy. Decedent data was acquired from the Los Angeles County Department of Medical Examiner-Coroner. We have added “demographic” to sentence three under “Intervention” in the “Methods” section to be more explicit.

3) The writing is very odd in parts of the paper. It really threw me that there are multiple instances of a patient "dispensing" a prescription which is not correct usage.

Response: We agree that “dispensing” is inaccurate when referring to patients, and have changed patient “dispensing” to patient “receiving” throughout the manuscript.

REVIEWERS' COMMENTS

Reviewer #1 (Remarks to the Author):

The authors did a fine job addressing the concerns raised by myself and the other reviewers to the original manuscript. I also appreciate the expanded discussion.

I was impressed to see how subtle the difference was between the standard and comparator letters: namely, a short paragraph near the end of the letter inviting recipients to keep the above content in mind, be comfortable voicing concerns and routinely logging into CURES. The context part of this “When your next patient presents with pain,” was just 7 words.

There are two comments:

- I find myself wondering whether some of the impact of the comparator letter comes from shifting the tone away from reflecting on the past bad outcome (and the associated guilt / negative affect) towards promotion of best practices in the future (with less bad affect).

- I also find myself wondering whether you would be able to measure whether those providers who received the comparator letter logged into the CURES system more. This would be beyond the scope of this paper, but the CA DOJ seems like it might make available this information identifying the clinicians usage patterns (without identifying the patients): <https://oag.ca.gov/cures/faqs#:~:text=In accordance with California Health,that may identify the patient%2C>.

Neither of these comments need to be addressed for publication of this manuscript.

Minor concern

- Method reference #5:

https://www.cdc.gov/drugoverdose/pdf/calculating_total_daily_dose-a.pdf no longer points to a valid website. Please update this appropriately.

Reviewer #2 (Remarks to the Author):

The revisions are acceptable. No new comments.

Reviewer #3 (Remarks to the Author):

Thank you for the thoughtful responses to my comments. The addition of the new benzodiazepine spillover analysis is interesting and adds another layer of depth to the paper. The only change that confused was the addition of these lines to the conclusions: "By concatenating these effects, we can infer that the letter with the If/when-then plan is roughly twice as effective as the letter in Doctor et al. 2018 relative to no treatment control." I think I know what the authors are saying, but it seems like a big stretch to just throw "twice as effective" out there with very crude back-of-the-envelope math. I would drop this or try to say something that is much more cleanly supported by the data in this study.

REVIEWERS' COMMENTS

Reviewer #1 (Remarks to the Author):

The authors did a fine job addressing the concerns raised by myself and the other reviewers to the original manuscript. I also appreciate the expanded discussion.

I was impressed to see how subtle the difference was between the standard and comparator letters: namely, a short paragraph near the end of the letter inviting recipients to keep the above content in mind, be comfortable voicing concerns and routinely logging into CURES. The context part of this “When your next patient presents with pain,” was just 7 words.

There are two comments:

- I find myself wondering whether some of the impact of the comparator letter comes from shifting the tone away from reflecting on the past bad outcome (and the associated guilt / negative affect) towards promotion of best practices in the future (with less bad affect).

- I also find myself wondering whether you would be able to measure whether those providers who received the comparator letter logged into the CURES system more. This would be beyond the scope of this paper, but the CA DOJ seems like it might make available this information identifying the clinicians usage patterns (without identifying the patients): <https://oag.ca.gov/cures/faqs#:~:text=In accordance with California Health,that may identify the patient%2C>.

Neither of these comments need to be addressed for publication of this manuscript.

Response: Thank you. We agree further research should elucidate whether positive, future-focused language is more effective than a negative emphasis on past experience. It would be interesting to test whether the comparator letter results in more self-monitoring behavior via CURES.

Minor concern

- Method reference #5: https://www.cdc.gov/drugoverdose/pdf/calculating_total_daily_dose-a.pdf no longer points to a valid website. Please update this appropriately.

Response: Thank you. We have removed this link.

Reviewer #2 (Remarks to the Author):

The revisions are acceptable. No new comments.

Reviewer #3 (Remarks to the Author):

Thank you for the thoughtful responses to my comments. The addition of the new benzodiazepine spillover analysis is interesting and adds another layer of depth to the paper. The only change that

confused was the addition of these lines to the conclusions: "By concatenating these effects, we can infer that the letter with the If/when-then plan is roughly twice as effective as the letter in Doctor et al. 2018 relative to no treatment control." I think I know what the authors are saying, but it seems like a big stretch to just throw "twice as effective" out there with very crude back-of-the-envelope math. I would drop this or try to say something that is much more cleanly supported by the data in this study.

Response: Thank you. We agree that our language was imprecise, and have changed the wording on page 9 of the Discussion.